# Development of bifunctional organocatalysts and application to asymmetric total synthesis of naucleofficine I and II

Yong-Hai Yuan[1], Xue Han[1], Fu-Ping Zhu[1], Jin-Miao Tian[2], Fu-Min Zhang[1], Xiao-Ming Zhang[1], Yong-Qiang Tu[1,2], Shao-Hua Wang [1] & Xiang Guo[1]

The proline-type organocatalysts has been efficiently employed to catalyze a wide range of asymmetric transformations; however, there are still many synthetically useful and challenging transformations that remain unachievable in an asymmetric fashion. Herein, a chiral bifunctional organocatalyst with a spirocyclic pyrrolidine backbone-derived containing fluoro-alkyl and aryl sulfonamide functionalities, are designed, prepared, and examined in the asymmetric Mannich/acylation/Wittig reaction sequence of 3,4-dihydro-$\beta$-carboline with acetaldehyde, acyl halides, and Wittig reagents. As a result, the spirocyclic pyrrolidine trifluoromethanesulfonamide catalyst can facilitate this versatile sequence as demonstrated by 18 examples displaying excellent enantioselectivity (up to 94% ee), as well as moderate to good yields (up to 54% over 3 steps). As a practical application, the asymmetric total synthesis of naucleofficine I (**1a**) and II (**1b**) in ten steps have been accomplished.

[1] State Key Laboratory of Applied Organic Chemistry and College of Chemistry and Chemical Engineering, Lanzhou University, Lanzhou 730000, P. R. China. [2] School of Chemistry and Chemical Engineering, Shanghai Jiao Tong University, Shanghai 200240, P. R. China. Correspondence and requests for materials should be addressed to Y.-Q.T. (email: tuyq@lzu.edu.cn)

The metal-free organocatalysis based on the dual activation of amino acids and their analogs has been displaying versatile utilities in the synthesis of natural products and pharmaceuticals[1–5]. The proline-type organocatalysts represent a large class of useful bifunctional catalytic systems, which have facilitated a wide range of asymmetric reactions of carbonyls, imines, and their variants, including aldol, Michael addition, Diels–Alder cycloaddition, Mannich reaction, and so on[6–12]. Despite these advances, there are still many inspiring transformations that remain unachievable in asymmetric synthesis. Therefore, development of more and reliable bifunctional organocatalysts is of high demand for effecting these transformations. Aimed at the asymmetric synthesis of biologically important natural products and drug molecules, our current efforts have been focused on developing novel and effective catalysis systems to enable synthetically challenging transformations so as to explore the concise synthetic approaches. As indicated in Fig. 1, at least four kinds of different natural products, the nauclecofficine I (1a) and II (1b), yohimbine, and berbanes, contain the common complex *N*-fused bicyclic structures 6 with multi-substituents and stereocenters. Compounds 1a and 1b are isolated from *Nauclea genus* and exhibit antibacterial and antiviral biological activities[13,14]. They represent the characteristic structures of a number of monoterpenoid indole alkaloids and have not been synthesized so far, although some approaches to their analogs have been reported by employing free radical cyclization and the Pictet–Spengler reaction as key steps[15–18].

In previous reports, the the spirocyclic pyrrolidine (SPD) backbone-derived organocatalysts have displayed high enantioselectivity in several reactions, possibly due to the rigid chiral environment of quaternary carbon stereocenter and spirocyclic scaffold, and the flexibility of the lone electron pair of the pyrrolidine moiety[19–21]. In this regard, it is hypothesized that attachment of the strong acidic fluoroalkylsulphonamide motif to a sterically hindered SPD backbone would generate a new bifunctional catalyst, which would exhibit stronger acidity and superior enantio-induction ability compared with proline but still maintain the active pyrrolidine amine. Furthermore, the BINOL-phosphonyl TfNH catalyst (Table 1, Cat 8) exhibits much stronger acidity than BINOL-phosphoric acid alone, due to its stability of counter anion and strong electron-accepting property of Tf group[22–25]. Herein, we present our research results on the asymmetric Mannich/acylation/Wittig sequence to efficiently assemble the tetrahydro-β-carboline 5, which allows us to accomplish the total synthesis of naucleofficine I (1a) and naucleofficine II (1b) in a concise way.

## Results

**Design plan.** The synthesis of natural products in Fig. 1 could be simplified to establishment of the single C3 stereocenter and installation of two unsaturated chains in 5. In this work, we focus on the synthesis of the alkaloids 1a and 1b, our retro-synthesis involves alternatively a late-stage *O*-hetero-Diels–Alder cycloaddition to construct the D/E rings with multi-functional groups and stereogenic centers in one step from the key intermediate 5 (Fig. 1). Therefore, we envision a versatile Mannich/acylation/Wittig sequence would enable the construction of 5 from commercially available 3,4-dihydro-β-carboline 2, acetaldehyde, acyl halides, and corresponding Wittig reagents. To achieve this catalytic asymmetric sequence, however, some challenges exist, including the following: (1) the starting *N*-aliphatic imine is relatively inactive compared with generally used active Boc-[26,27], Ts-[28,29], aryl-imine species[30,31], and Cbz-[32], which requires a stronger acidity to activate the C=N bond via hydrogen bonding; (2) the active acetaldehyde[33–39] could easily form byproducts via aldol condensation; and (3) asymmetric Mannich reaction of 3,4-dihydro-β-carboline and aldehydes is a challenging work, which has not been reported up to now, maybe due to the instability of the resulting amino aldehyde.

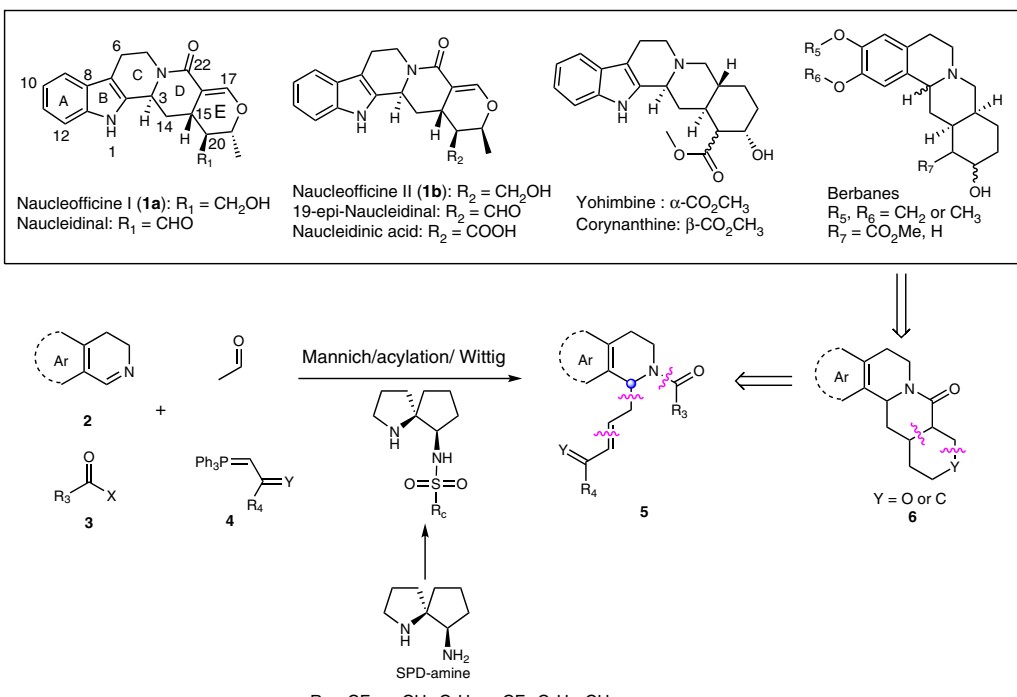

**Fig. 1** Design of asymmetric synthesis of monoterpenoid alkaloids. Retro-synthesis involves Diels–Alder reaction and Mannich/acylation/Wittig sequent reaction

**Table 1 Optimization of the Mannich/Acylation/Wittig reaction conditions[a]**

| Entry | Catalyst | T [ºC] | Additive | t [h] | Yield [%][b] | ee [%][c] |
|---|---|---|---|---|---|---|
| 1 | Cat 1 | 0 | - | 8 | 21 | 51 |
| 2 | Cat 1 | 0 | $PhCO_2H$ | 8 | 8 | −1.1 |
| 3 | Cat 1 | 0 | PTS | 8 | 12 | −57 |
| **4** | **Cat 1** | **0** | **Et₃N** | **4** | **51** | **91** |
| 5 | Cat 1 | 0 | DMAP | 20 | 29 | 87 |
| 6 | Cat 1 | 0 | DBU | 20 | 47 | 86 |
| 7 | Cat 1 | −5 | Et₃N | 10 | 48 | 88 |
| 8 | Cat 1 | −10 | Et₃N | 10 | 45 | 89 |
| 9 | Cat 2 | 0 | Et₃N | 48 | 7 | 3 |
| 10 | Cat 3 | 0 | Et₃N | 48 | 12 | 8 |
| 11 | Cat 4 | 0 | Et₃N | 48 | <5 | 32 |
| 12 | Cat 5 | 0 | Et₃N | 48 | <5 | 18 |
| 13 | Cat 6 | 0 | Et₃N | 48 | <5 | 14 |
| 14 | Cat 7 | 0 | Et₃N | 48 | ND[d] | ND[d] |
| 15 | Cat 8 | 0 | Et₃N | 30 | 15 | 20 |
| 16 | Cat 9 | 0 | Et₃N | 6 | 26 | 68 |
| 17 | Cat 10 | 0 | Et₃N | 48 | 20 | 69 |
| 18 | Cat 11 | 0 | Et₃N | 48 | <5 | 67 |
| 19 | Cat 12 | 0 | Et₃N | 48 | 6 | 20 |
| 20 | Cat 13 | 0 | Et₃N | 48 | 9 | 29 |

[a] Unless otherwise noted, all reactions were carried with **2a** (0.1 mmol), acetaldehyde (0.3 mmol), catalyst (0.02 mmol), and additive (0.05 mmol), in 0.5 mL DCE and 0.5 mL $H_2O$; after the reaction is complete, the solution was evaporated, then DCM (1 mL), $K_2CO_3$ (0.2 mmol), acyl halide **3** (0.15 mmol), and Wittig reagent **4** (0.2 mmol) was added sequentially to the reaction system
[b] Isolated yield over three steps
[c] Determined by HPLC analysis
[d] No reaction detectable
*DBU* 1,8-diazabicyclo[5.4.0]undec-7-ene, *DMAP* 4-dimethylaminopyridine, *PTS* p-toluenesulfonic acid, *DCE* 1,2-dichloroethane, *DCM* dichloromethane
Bold values represent the optimal condition

**Reaction condition optimization.** Initially, four catalysts **Cat 1–4** with substituents possessing different electronic properties were prepared from (S,S)-SPD-amine (for details, see Supplementary Note 2). Imine **2a** was selected as a model substrate to screen the optimal condition of the Mannich sequence with acetaldehyde, acetyl bromide, and methyl(triphenylphosphoranylidene)-acetate. Fortunately, the expected Mannich sequence could take place in the presence of **Cat 1** in DCE/$H_2O$ at 0 °C, affording mainly the desired product 5a albeit with the poor yield (21% over three steps) and moderate enantiomeric excesses (ee) (51%) (Table 1, entry 1). Encouraged by this result, the effect of additives was then investigated (for details, see Supplementary Table 1). The presence of a basic additive was found to be crucial to improve the reaction outcome, with Et₃N giving the best result (entry 4), the main byproducts were the decomposition of the amino aldehyde in the second step of acylation and the byproducts of aldol condensation have not been detected. In contrast, the acid additives gave poor results and reversed the chirality of the resulting products (entries 2–3 vs. 4–6). Subsequently, decreasing the reaction temperature led to slightly lower yield and ee (entries 7–8). Further examination of the analogous **Cat 2–4** revealed that the TfNH modified **Cat 1** provided a much better result than **Cat 2–4** with the weaker and stronger electron withdrawing property of tolylsulfonyl, p-(triflouromethyl)-benzenesulfonyl, and methylsulfonyl, respectively (entries 9–11 vs. 4). In addition, other types of catalysts

with quite different levels of reactivity and/or stereo-effect from **Cat 1** were subjected to the reaction and compared with **Cat 1** under the same condition as entry 4. These include the following: (1) **Cat 5**, **Cat 13**, which does not have an acidic functional group; (2) **Cat 6**, which has a protecting group of pyrrolidine amine; (3) **Cat 8**, which has a single, stronger acidity and generally identified excellent enatio-induction of BINOL framework; and (4) **Cat 9–12**[5,40–44], which have similar reactivity but quite different stereo-effect of a linear derivative. Unfortunately, all of these catalysts gave poor results (entries 12–18). Particularly, **Cat 7** failed to promote this sequence and only the starting material was recovered (entry 14). These results indicated that **Cat 1** exhibited both excellent reactivity and high enantio-induction ability toward the studied transformation.

**Substrate scope.** Subsequently, the optimal condition of entry 4 (the bold values in table 1) was employed for substrate expansion. First, a variety of substituted 3,4-dihydro-β-carbolines **2b–2j** with a variety of substituents on the indole rings were examined and all reacted smoothly to afford the desired products **5b–5j** (Fig. 2a)[45–49]. In detail, the EWG (electron-withdrawing group) substitutions with F, Cl, and Br at C9–C11 of the indole ring generally gave high ee (90–92%) and good yields (51–54% over three steps) (**5d–5h**), whereas the EDG (electron-donating group) substitutions gave slightly lower ee (88% ee) and yields (45%) (**5b** and **5c**). Interestingly, the C12 substitutions with either EWG (**5i**) or EDG (**5j**) both gave high ee (94% and 90% ee) but merely moderate yields (43% and 37%). Moreover, substitutions at the indole nitrogen atom with the EWG (**5k**) or EDG (**5l**) resulted in similarly lower ee (83% and 76%) and varied yields (52% and 32%). In addition, variation of acyl halides could also generate high ee (both 91% ee) and moderate yields (45% and 46%) (**5m** and **5n**), respectively. Moreover, changing the Wittig reagents gave moderate yields (42% and 40%) and high ee (both 91% ee) (**5o** and **5p**), which were the key intermediates for the synthesis of natural products **1a/1b**. Notably, two 3,4-dihydro-isoquinoline substrates were explored for the catalytic Mannich/ acylation/Wittig sequence, which also proved to be effective, generating the corresponding products **5q** and **5r** with good ee (81% and 87%) and moderate yields (41% and 36%), respectively (Fig. 2b). As a result, the explored **Cat 1** could facilitate well the expected asymmetric Mannich sequence with a wide substrate toleration in good to high efficiency in most cases.

**Asymmetric total synthesis of naucleofficine I and II.** To demonstrate the utility of this sequence, the asymmetric total synthesis of **1a/1b** were carried out efficiently from intermediates **5o**. First, **5p** underwent an *O*-hetero-Diels–Alder cycloaddition under reflux in mesitylene to produce the advanced pentacyclic intermediate **7** with mainly *cis*-configuration of C3 and C15 (Fig. 3)[50]. Next, we turned to the asymmetric total synthesis of the natural alkaloids **1a** and **1b** from intermediate **5o**, which was readily prepared on the gram-scale from **2a** using 10 mol % (*R,R*)-**Cat 1** in 42% yield and 91% ee (Fig. 4). After extensive screening of the reaction conditions (for details, see Supplementary Table 2), ZnBr$_2$ was found to enable the expected *O*-hetero-Diels–Alder cyclization of **5o**, efficiently affording the pentacyclic intermediate **8** with four desired stereocenters in 51% yield and >20:1 dr (Fig. 5), and there were amount of natural products that owned *trans*-configuration of C3 and C15 had not been synthesized. One-recrystallization of **8** in MeOH/Et$_2$O enhanced the enantio-purity to >99% ee (85% yield) and its absolute configuration was confirmed by X-ray crystallographic analysis (for details, see Supplementary Table 6). However, a thermal

cyclization of **5o** in mesitylene under refluxing temperature resulted mainly in the formation of undesired C15-epimer. Hydroformylation of **8** with POCl$_3$/DMF afforded aldehyde **9** in 48% yield, which was treated with freshly prepared Me$_2$CuLi to provide the separable products **10**, **11**, and **11′** (2.8: 2.5: 1) with 63% total yield. Fortunately, isomer **11′** could be readily converted to **11** with 85% yield when stirred in DBU (1,8-diazabicyclo[5.4.0]undec-7-ene)/DCM. Subsequent reduction of **10** with NaBH$_4$ furnished **12** with 86% yield. After extensive investigations[51,52], protection of the hydroxyl group of **12** with hexamethyl disilylamine at refluxing temperature gave a silyl ether, which was subjected to elimination of methoxy group and removal of Boc with BF$_3$·Et$_2$O and TMSOTf in CH$_3$CN to obtain acetal **14**, and then the double bond isomerization to give **1a** with 81% yield, so we developed an efficient method to construct the double bond of these natural products. In the same way, **1b** was also synthesized separately from **11**. The spectra data of synthetic (+)-**1a** and (−)-**1b** are consistent with literature[13], and the ee of (+)-**1a** and (−)-**1b** are >99% determined by chiral high-performance liquid chromatography (HPLC).

## Discussion

In summary, a chiral bifunctional organocatalyst has been successfully developed, which is a good assembly of strong acid and sterically hindered SPD framework, and thus exhibits high reactivity as well as excellent stereo-induction ability toward the asymmetric Mannich reaction of 3,4-dihydro-β-carboline with a broad substrate toleration. Furthermore, synthetic utility of this Mannich sequence has been exemplified through the asymmetric total synthesis of naucleofficine I (**1a**) and II (**1b**) in a concise way. Further expanded studies on asymmetric catalysis with our bifunctional organocatalysts toward other transformations and synthesis of other natural products are underway.

## Methods

**General information.** All reactions requiring anhydrous conditions were carried out under argon atmosphere using oven-dried glassware (130 °C), which was cooled under argon. All solvents were purified and dried by standard techniques, and distilled prior to use. All reactions under standard conditions were monitored by thin-layer chromatography (TLC) on gel F$_{254}$ plates. The products were purified by flash column chromatography on silica gel (200~300 mesh) or neutral alumina (200~300 mesh). $^1$H NMR, $^{13}$C NMR, and $^{19}$F NMR spectra were obtained on Bruker AM-400, JEOL JNM-ECS-400, or Varian Mercury-600. Chemical shifts (δ) were reported in p.p.m. relative to residual solvent signals (CDCl$_3$: 7.26 p.p.m. for $^1$H NMR, 77.0 p.p.m. for $^{13}$C NMR; DMSO-$d_6$: 2.50 p.p.m. for $^1$H NMR, 39.5 p.p. m. for $^{13}$C NMR; CD$_3$OD: 3.31 p.p.m. for $^1$H NMR, 49.0 p.p.m. for $^{13}$C NMR). The following abbreviations were used to indicate the multiplicity in NMR spectra: s, singlet; d, doublet; t, triplet; m, multiplet. High-resolution mass spectral analysis data were measured on the Bruker ApexII with ESI resource. Infrared spectra were recorded on Nicolet FT-170SX spectrometer. Melting points were measured on a melting point apparatus and were uncorrected. The ee of the products were determined by HPLC analysis. X-ray diffraction data were collected on Agilent SuperNova Eos diffractometer. Optical rotations were detected on RUDOLPH A21202-J APTV/GW.

**General procedure for the Mannich sequent reaction.** To a solution of substituted 3,4-dihydro-β-carboline **2** (0.1 mmol), **Cat 1** (5.4 mg, 0.02 mmol) in DCE/H$_2$O (1 mL, *v/v* 1: 1) were added Et$_3$N (7 μL, 0.05 mmol) and aldehyde (16 μL, 0.3 mmol) at 0 °C. The mixture was stirred at this temperature until the substrate disappeared via TLC detection. Then the resulting solution was concentrated in vacuo. Then 1 mL DCM, K$_2$CO$_3$ (27.6 mg, 0.2 mmol), and acetyl bromide **3a** (10 μL, 0.12 mmol) were added at 0 °C sequentially. About 30 min later, **4a** (66.8 mg, 0.2 mmol) was added to the reaction at room temperature and reacted at the same temperature for 12 h. The mixture was then quenched with water and the mixture was extracted with DCM. Then the combined organic phase was washed with saturated brine, dried over anhydrous Na$_2$SO$_4$, and concentrated in vacuo. The residue was purified through column chromatography on neutral alumina to give substrates **5**.

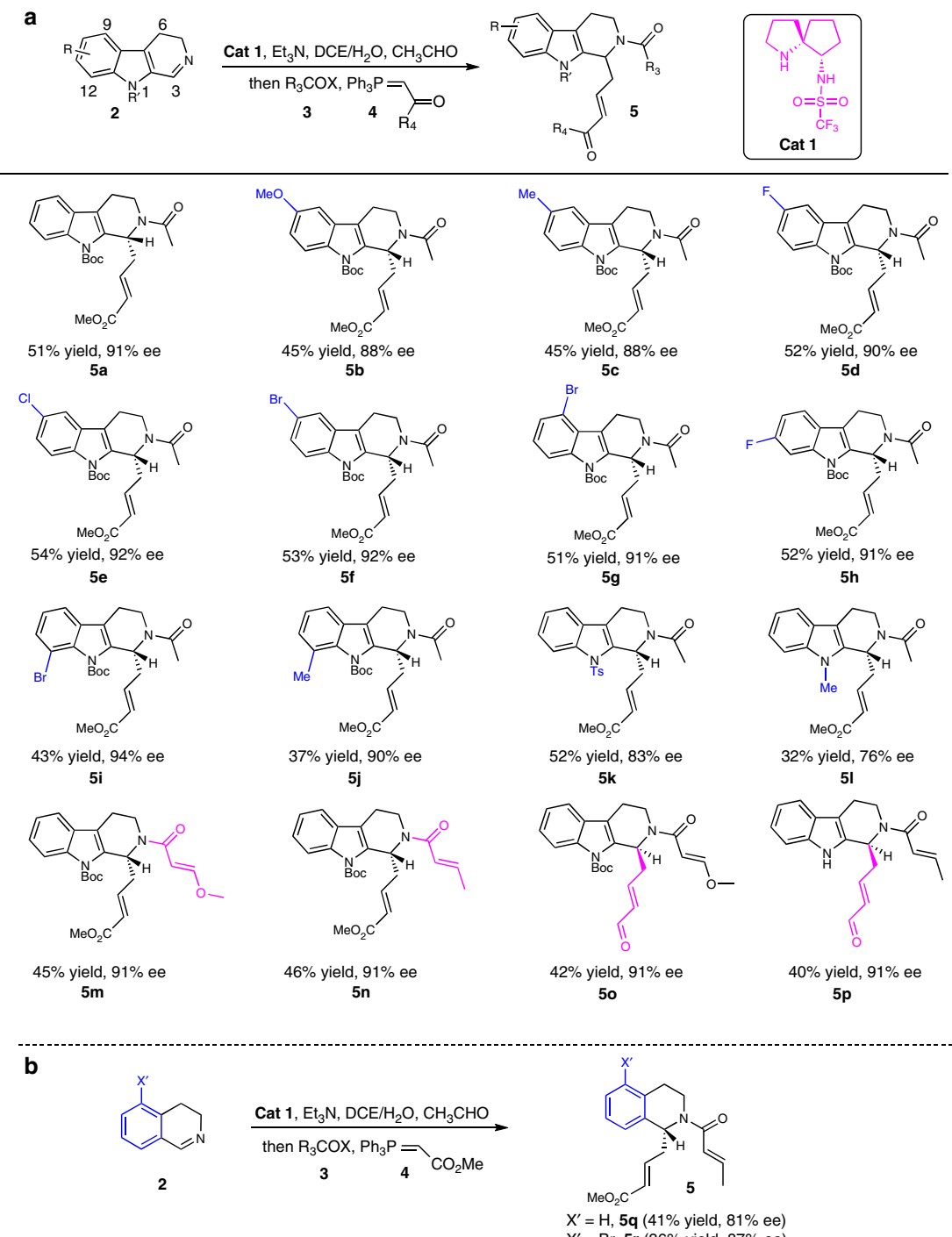

**Fig. 2** The substrate scope of the Mannich/Acylation/Wittig reaction. Reaction condition: **2a** (0.1 mmol), acetaldehyde (0.3 mmol), (*S,S*)-**Cat 1** (0.02 mmol), and Et₃N (0.05 mmol), in 0.5 mL DCE and 0.5 mL H₂O, then acyl halide **3** (0.15 mmol) and Wittig reagent **4** (0.2 mmol). Isolated yield over three steps. Enantiomeric excess determined by HPLC analysis. **a** The scope of 3,4-dihydro-β-carboline substrates. **b** The scope of 3,4-dihydro-isoquinoline substrates

**Fig. 3** Further transformation. Thermocatalytic hetero-Diels–Alder reaction mainly obtained the *cis*-product of **C3** and **C15**

**Fig. 4** Gram-scale synthesis of **5o**. Reactions were performed with 10% mol (*R,R*)-**Cat 1**

**Fig. 5** Asymmetric total synthesis of naucleofficine I and II. The key steps include *O*-hetero-Diels–Alder cycloaddition, hydroformylation, and double bond isomerization

## Data availability

The X-ray crystallographic coordinates for structures reported in this study have been deposited at the Cambridge Crystallographic Data Centre (**Cat 1**: CCDC 1875257, **8**: CCDC 1875256). These data can be obtained free of charge from The Cambridge Crystallographic Data Centre via www.ccdc.cam.ac.uk/data_request/cif.

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

## Acknowledgements

We thank the National Natural Science Foundation of China (Nos. 21502080, 21772071, 21871117, 21772076), National Science and Technology Major project of the Ministry of Science and Technology of China (2018ZX09711001-005-002), and the '111' Program of MOE for their financial support.

## Author contributions

Y.-H.Y. performed all of experiments and prepared the supplementary information. X.H. and X.G. prepared materials for the synthesis of naucleofficine I and II. X.H. prepared substrates for substrate expansion. F.-P.Z. synthesized **Cat 1**. J.-M.T. helped the design of Mannich sequent reaction. Y.-Q.T., F.-M.Z., X.-M.Z. and S.-H.W. directed the project and discussed the results. Y.-Q.T. wrote the manuscript.

## Additional information

**Competing interests:** The authors declare no competing interests.

7