## [Peer Review File · Nature Communications]

Reviewers' comments:

Reviewer #1 (Remarks to the Author):

Manuscript Title: Development of Bifunctional Organocatalysts and Application to Asymmetric Total Synthesis of Naucleofficine I and II

Authors: Yong-Hai Yuan, Xue Han, Fu-Ping Zhu, Jin-Miao Tian, Fu-Min Zhang, Xiao-Ming Zhang, Yong-Qiang Tu, Shao-Hua Wang & Xiang Guo

Manuscript Number: NCOMMS-19-12726

Recommendation: The paper may be accepted, however, after a major revision as noted below.

Comments to the Authors:

Yuan and co-workers have discovered a novel chiral bifunctional organocatalyst for enamine based catalysis, which is an assembly of strong acid and sterically hindered spirocyclic pyrrolidine framework. This catalyst exhibits high reactivity as well as excellent stereoselectivity toward the Mannich/acylation/Wittig sequence of 3,4-dihydro- β -carboline, which sets a chiral center in enantioselective fashion. Interestingly, synthetic utility of this Mannich sequence has been highlighted the first asymmetric total synthesis of naucleofficine I (1a) and II (1b). There are two components in this article namely development of new organocatalyst (under enamine catalysis) as well as application. The research work has been executed very well and all new compounds have been fully characterized by their spectroscopic data, but there is some concern which must be taken care before publishing.

The concern is authors didn't utilize Jorgensen & Hayashi's catalyst in their optimization (see; (Hayashi et. al. *Org. Lett.* 2009, 11, 3854; this reference is missing). This catalyst is found to be universal for catalytic enantioselective reactions through enamine/dienamine/trienamine catalysis. Therefore, authors must utilize these catalysts and compare their newly designed catalyst with the existing ones. So far as total syntheses of naucleofficine I (1a) and II (1b) is concerned, authors have utilized a key O-hetero-Diels-Alder cycloaddition reaction, which sets all the rings required for these alkaloids, followed by some synthetic manipulations. The total syntheses have been achieved in 6

steps from compound 5o, which is quite impressive. Therefore, this manuscript has the potential to be accepted if authors address above concerns.

Reviewer #2 (Remarks to the Author):

This manuscript reported the development of a new type of bifunctional organocatalyst, which promoted enantioselective Mannich reaction of unactivated cyclic N-aliphatic imines in a highly enantioselective manner. Although a combination of pyrrolidine motif and acidic sulphonamide motif, such as Cat.9, were previously developed as the bifunctional organocatalyst, and SPD-derived organocatalysts have been reported, Cat 1-4 in this manuscript have not been reported so far. As shown in Table 1, Cat 1 promoted the reaction with much higher ee as compared with the known well-established bifunctional organocatalysts, indicating a high potential of SPD-derived amino sulphonamide bifunctional catalysts, and Cat 1 provides another option for enantioselective organocatalysis.

Substrate generality (functional compatibility) of this catalysis was rather limited, but the application to short-step syntheses of nucleofficines are very interesting, though diastereoselectivity in methylation step was very low.

Coupled with the importance of the obtained chiral products, this reviewer recommends this manuscript for publication in Nature Communications.

Before publication, however, the following issues should be addressed properly.

(1) The authors mentioned the total synthesis of nucleofficines was achieved in 6 steps, but it is not correct. The starting 2a was synthesized from 2,3,4,9-tetrahydro-1H-Pyrido[3,4-b]indole in 2 steps, and the key process from 2a to 5o is one-pot 3 steps sequential reactions. So, it should be overall 10 steps total synthesis from commercially available 2,3,4,9-tetrahydro-1H-Pyrido[3,4-b]indole.

(2) An alternative approach to 5 would be (1) acylation of 2a to generate more reactive acyl iminium cation, (2) enantioselective Mannich reaction, and (3) Wittig reaction. In this case, we can use more reactive substrate for enantioselective Mannich reaction. In fact, Mannich type reaction of 2-acyltetrahydroisoquinolines was reported (Synthetic Communications, 26(11), 2135-44; 1996). Have the authors examined this type of alternative approach?

(3) (Table 1) "Ph" is equivalent to "C₆H₅" not "C₆H₄". Therefore, it would be better to use "p-CH₃-C₆H₄" rather than "p-CH₃-Ph.

(4) (Table 2) In some compounds, such as 5e, CO₂Me was centered. It should be right-aligned.

The Response to Reviewers

For Reviewer #1 :

- 1.1) The concern is authors didn't utilize Jorgensen & Hayashi's catalyst in their optimization (see; (Hayashi et. al. Org. Lett. 2009, 11, 3854; this reference is missing). This catalyst is found to be universal for catalytic enantioselective reactions through enamine/dienamine/trienemine catalysis. Therefore, authors must utilize these catalysts and compare their newly designed catalyst with the existing ones.

Reponse : Thank you very much for your valuable suggestion.

We applied Jorgensen & Hayashi's catalysts (Cat 12, Cat 13) to our asymmetric Mannich/ acylation/ Wittig sequent reaction under the optimal reaction conditions, and the results were listed in the Table R1. Compared with our newly designed catalyst **Cat 1**, these catalysts gave unsatisfactory results, and mainly the starting material was recovered. The results had been added in the updated supporting information (S13). The reference (Org. Lett. 2009, 11, 3854) has been added as ref. 32 in the revised manuscript, and the corresponding number of references after ref. 32 should also be changed.

Table R1 The comparison reaction results of three catalysts

entry	Cat	t/h	Yield (%)	ee (%)
1	Cat 1	4	51	91
2	Cat 12	48	6	20
3	Cat 13	48	9	29

For Reviewer #2 :

- 2.1) The authors mentioned the total synthesis of nucleofficines was achieved in 6 steps, but it is not correct. The starting **2a** was synthesized from 2,3,4,9-tetrahydro-1H-

Pyrido-[3,4-b]-indole in 2 steps, and the key process from **2a** to **5o** is one-pot 3 steps sequential reactions. So, it should be overall 10 steps total synthesis from commercially available 2,3,4,9-tetrahydro-1H-Pyrido[3,4-b]indole.

Response : Thank you very much for your valuable suggestion.

We have changed “6 steps” to “10 steps” in the revised manuscript.

2.2) An alternative approach to **5** would be (1) acylation of **2a** to generate more reactive acyl iminium cation, (2) enantioselective Mannich reaction, and (3) Wittig reaction. In this case, we can use more reactive substrate for enantioselective Mannich reaction. In fact, Mannich type reaction of 2-acyltetrahydroisoquinolines was reported (Synthetic Communications, 26(11), 2135-44; 1996). Have the authors examined this type of alternative approach?

Response : Thank you very much for your valuable suggestion.

According to referee's suggestion, we have tried to activate the imine by initial formation of acyl iminium cation follow by Mannich and Wittig type reaction (Reaction conditions: reactions were carried out with **2a** (0.1 mmol), crotonoyl chloride (0.15 mmol) in 1 mL DCE, then acetaldehyde (0.15 mmol), catalyst (0.02 mmol) were added until the starting materials disappeared via TLC detection. After reaction is complete, Wittig reagent **4** (0.2 mmol) was added sequentially to the reaction system). The reaction proceeded smoothly (yield = 25%), albeit with the very poor enantioselectivity (ee <10%). The corresponding detail has been added in the updated supporting information (S 14).

2.3) (Table 1) “Ph” is equivalent to “C₆H₅” not “C₆H₄”. Therefore, it would be better to use “p-CH₃-C₆H₄” rather than “p-CH₃-Ph”.

Response : Thank you very much for your valuable suggestion.

We have revised “p-CH₃-Ph, p-CF₃-Ph” to “p-CH₃-C₆H₄, p-CF₃-C₆H₄” in Figure 1 and Table 1.

2.4) (Table 2) In some compounds, such as **5e**, CO₂Me was centered. It should be rightaligned.

Response : Thank you very much for your valuable suggestion.

We have revised CO₂Me of **5e**, **5h**, **5k**, **5n**.

REVIEWERS' COMMENTS:

Reviewer #1 (Remarks to the Author):

Manuscript Title: Development of Bifunctional Organocatalysts and Application to Asymmetric Total Synthesis of Naucleofficine I and II

Authors: Yong-Hai Yuan, Xue Han, Fu-Ping Zhu, Jin-Miao Tian, Fu-Min Zhang, Xiao-Ming Zhang, Yong Qiang Tu, Shao-Hua Wang & Xiang Guo

Reviewer's Report: In this report a novel chiral bifunctional organocatalyst having a strong acid and sterically hindered SPD framework has been successfully developed, and utilized in Mannich/acylation/Wittig sequent reaction of 3,4-dihydro- β -carboline with a broad substrate toleration. This newly developed catalyst exhibits high reactivity as well as excellent stereoinduction ability toward the Mannich/acylation/Wittig sequence.

I can see that authors have done modifications based on the reviewer's comments. However, I insist that results from Jorgensen and Hayashi's catalyst (which is given in SI, S-13 page) should be given in the manuscript itself, as these catalysts are universal and cater a vast variety of organic transformations.

Another major point that, authors have claimed the newly developed catalyst catalyses Mannich/acylation/Wittig sequent reaction of 3,4-dihydro- β -carboline with a broad substrate toleration. However, this is not true. It catalyses only Mannich reaction, which is crucial for the Mannich/acylation/Wittig sequence. In fact, acylation/Wittig sequence doesn't require organocatalyst with SPD framework. I suggest authors to take care of this in the revised manuscript.

Reviewer #2 (Remarks to the Author):

This revised manuscript has made appropriate modifications based on referees' comments.

So now, this reviewer thinks this manuscript is suitable for the publication in Nature Communications.

The Response to Reviewers

For Reviewer #1:

- 1.1) I can see that authors have done modifications based on the reviewer's comments. However, I insist that results from Jorgensen and Hayashi's catalyst (which is given in SI, S-13 page) should be given in the manuscript itself, as these catalysts are universal and cater a vast variety of organic transformations.

Response: *Thank you very much for your valuable suggestion.*

Cat 12 and Cat 13 have been placed in the updated manuscript (see Table 1). For the more details, also please see the revised manuscript.

- 2.2) It catalyses only Mannich reaction, which is crucial for the Mannich/acylation/Wittig sequence. In fact, acylation/Wittig sequence doesn't require organocatalyst with SPD framework. I suggest authors to take care of this in the revised manuscript.

Response: *Thank you very much for your valuable suggestion.*

We have changed "Mannich/ acylation/ Wittig sequent" to "asymnetric Mannich" in the Discussion section of the revised manuscript.

For Reviewer #2:

- 3.1) This revised manuscript has made appropriate modifications based on referees' comments. So now, this reviewer thinks this manuscript is suitable for the publication in Nature Communications.

Response: *Thank you very much for your positive comments.*